# Detection of Infectious *Cryptosporidium parvum* Oocysts from Lamb’s Lettuce: CC–qPCR’s Intake

**DOI:** 10.3390/microorganisms9020215

**Published:** 2021-01-21

**Authors:** Sophie Kubina, Damien Costa, Loïc Favennec, Gilles Gargala, Angélique Rousseau, Isabelle Villena, Stéphanie La Carbona, Romy Razakandrainibe

**Affiliations:** 1ACTALIA Food Safety Department, 310 Rue Popielujko, 50 000 Saint-Lô, France; angelique.rousseau@live.com (A.R.); s.lacarbona@actalia.eu (S.L.C.); 2Laboratoire de Parasitologie, EA 7510, Université de Rouen Normandie, 76 000 Rouen, France; damien.costa@chu-rouen.fr (D.C.); loic.favennec@univ-rouen.fr (L.F.); gilles.gargala@chu-rouen.fr (G.G.); 3Laboratoire de Parasitologie-Mycologie, CNR Laboratoire Expert Cryptosporidioses, Centre Hospitalier Universitaire de Rouen, 76 000 Rouen, France; 4Laboratoire de Parasitologie-Mycologie, EA 7510, SFR CAP-Santé, Université Reims-Champagne Ardenne, Centre Hospitalier Universitaire de Reims, 51 000 Reims, France; ivillena@chu-reims.fr

**Keywords:** *Cryptosporidium*, oocysts infectivity, detection, fresh vegetables, in vitro method

## Abstract

*Cryptosporidium* spp. is responsible for several food and waterborne disease outbreaks worldwide. Healthier lifestyles attract consumers to eat, notably, fresh food like fruits and vegetables. The consumption of raw or under-cooked food increases the risk of foodborne transmission of Cryptosporidiosis. The assessment of the consumer’s exposure to *Cryptosporidium* danger is crucial for public health. Still, the standardized method to detect this parasite in fresh leafy greens and berry fruits has only been available since 2016 and suffers from weaknesses. Consequently, in this study, we propose a method with minimum processing steps that combines cell culture and the quantitative PCR (CC–qPCR) for detecting infectious *C. parvum* oocysts recovered from lamb’s lettuce. This CC–qPCR is a rapid and easy method that can detect up to one oocyst, whereas it is undetectable by classic qPCR.

## 1. Introduction

During the last decade, food choices and eating habits have changed dramatically around the world and contributed to the globalization of food supplies. This trend has been implicated as a reason for the emergence or re-emergence of many foodborne parasitic diseases [1]. The consumption of raw or under-cooked foods and the growing market for more ready-to-eat fresh and healthy food, as well as novel, ethnic food products may be linked to the increased risk of foodborne transmission of zoonoses. Fruits, vegetables, and drinking water have been implicated as vehicles for many environmental stages of parasites, particularly protozoa such as *Cryptosporidium*, *Giardia*, and *Cyclospora* [2].

In Europe, of 279 parasitic protozoan outbreaks in humans (representing 40,289 cases) reported since 2010 to the European Food Safety Authority (EFSA) and the European Centre for Disease Prevention and Control, 58 were due to *Cryptosporidium* spp., representing 33,486 cases. Currently, *Cryptosporidium* is considered one of the most important etiological agents of food and waterborne disease [3,4,5,6,7,8,9,10,11,12,13]. However, the number of foodborne parasitic outbreaks is probably underestimated. Indeed, in 32% of reported outbreaks, the pathogen was not identified. Additionally, the methods for its detection are not suitable for all food matrices and were only recently published [14]. Among foodborne *Cryptosporidium* outbreaks described around the world, salad consumption was incriminated in 35% of cases [15]. Lamb’s lettuce is usually included in a pre-packed mixed salad. Moreover, lamb’s lettuce grows low to the ground and is entirely consumed (no leaves are thrown out), increasing both the likelihood of surface contamination and the transmission of the infective stages of the protozoa to humans [16].

Since 2016, a gold standard exists for *Cryptosporidium* oocyst detection in food (ISO 18744:2016), but it suffers from weaknesses such as a relatively low parasite recovery rate (34%) that is variable between users [14]. Indeed, (i) the small number of organisms relative to the amount of the interfering material and (ii) the specific characteristics for each food matrix (trapping and adhesion force) may interfere with oocyst removal and elution [17,18]. Furthermore, the standard ISO method is neither able to determine the species/genotypes of the isolated oocysts nor to presume the infectivity of the oocysts [19]. Acquiring such data is crucial for assessing and identifying the risk of *Cryptosporidium* infection from consumption of contaminated food. Methods based on cell culture coupled to quantitative PCR (CC–qPCR) are interesting tools to evaluate the infectivity of *Cryptosporidium* oocysts [20]. Cell culture assays are based on the in vitro release of sporozoites from oocysts after induction of excystation [21]. Sporozoites invade epithelial cells and start their life cycle; specific detection of *C. parvum* (Tyzzer, 2012) DNA is thus possible when combining cell culture with the qPCR technique and using specific primers and probes [22,23]. Such assays have already been used to detect infectious oocysts in water and wastewater but were never applied to complex food matrices [20]. In addition to the assessment of infectivity, the in vitro proliferation of the parasite allows an increase in DNA quantities that could enhance the detection of a small number of oocysts in the sample and, consequently, improve the detection compared to other direct molecular assays.

This study aims to propose a CC–qPCR method with minimum processing steps to detect the *C. parvum* infectious oocysts recovered from lamb’s lettuce. Several publications recommend oocyst excystation for improving in vitro parasitic proliferation. However, the aim is also to limit the supplementary loss of parasites during excystation treatment. Thus, on one hand, the human ileocecal adenocarcinoma cell line (HCT-8) was inoculated with purified or excysted oocysts, based on known procedures; on the other hand, the method detection limit was assessed by inoculating the cell culture directly with known concentration ranges of purified parasites, and finally with concentration ranges of recovered oocysts from lamb’s lettuce.

## 2. Materials and Methods

### 2.1. Cryptosporidium parvum Oocysts and Purification

*Cryptosporidium parvum* oocysts were purchased from the National Institute of Agricultural Research (INRA), Nouzilly, France. The isolate was initially sampled from an infected child and maintained in a laboratory by serial passages on calves at the INRA, Nouzilly, France. Several batches of parasites were used in this study. Oocysts in the feces of an experimentally infected calf were concentrated and purified using a commercially available kit (Isolate for ImmunoMagnetic Separation “IMS” of *Cryptosporidium* oocysts, TCS biosciences, Botolph Claydon, UK) according to the manufacturer’s protocol. Purified oocysts were enumerated in triplicate by a Kova-slide hemocytometer (Hycor, Garden Grove, CA, USA) and were stored at +4 °C in sterile phosphate-buffered saline (PBS) solution until use.

### 2.2. Cell Line and Routine Cell Culture

The human ileocecal adenocarcinoma HCT-8 cell line (ATCC CCL-224) was maintained in Rosewell Park Memorial Institute “RPMI“1640 medium (Lonza, Verviers, Belgium) supplemented with 5% heat-inactivated fetal bovine serum (Eurobio, Les Ulis, France), 100 UI/mL of penicillin (Corning, Manassas, VA, USA), and 100 pg/mL of streptomycin (Corning, Manassas, VA, USA) in Falcon flasks (75 cm^2^) at 37 °C in a 5% CO_2_ atmosphere. HCT-8 cells were passaged every three to four days. Cells were transferred to twenty-four-well plates (Thermo Fischer Scientific, Roskilde, Denmark) and grown to 90% confluent monolayers in a 5% CO_2_ humidified incubator before infection.

### 2.3. Infection of Cell Monolayers

HCT-8 cells were seeded at a density of 2 × 10^4^ cells per well in 24-well flat-bottom plates. Monolayers were allowed to grow at 37 °C in a 5% CO_2_ atmosphere and to reach 90% confluence before the infectivity assay. Excysted or not, oocysts (10^4^ per well) were inoculated onto monolayers. During this study, the excystation procedure was adapted from those described elsewhere such as (i) 1.5% sodium taurocholate (NaTC; Sigma, Steinheim, Germany) solution in RPMI 1640 medium at 37 °C for 1 h [21]; (ii) 2.63% aqueous sodium hypochlorite at room temperature for 10 min [18]; and (iii) the combination of 2.63% aqueous sodium hypochlorite at room temperature for 10 min, then centrifugation at 15,000× *g* for 10 min, and 1.5% NaTC solution in RPMI 1640 medium at 37 °C for 1 h. After the excystation, the parasitic suspension (oocysts and sporozoites) was centrifugated at 15,000× *g* for 10 min and resuspended in a final volume of 1 mL in co-culture medium (RPMI 1640 medium supplemented with 45 mg/mL of glucose (Sigma), 40 μg/mL of para-aminobenzoic acid (Sigma), 40 μg/mL of 2-[4-(2-hydroxyethyl)piperazin-1-yl]ethanesulfonic acid “HEPES” (Sigma), 0.35 mg/mL of ascorbic acid (Sigma), 1 UI/mL of insulin (Sigma), 100 UI/mL of penicillin (Panpharma, Luitré-Dompierre, France), and 100 pg/mL streptomycin (Panpharma) before inoculation onto the monolayer. Parasitic suspensions were deposited in flat-bottom 24-well plates with and without cells in a co-culture medium and incubated at 37 °C in a 5% CO_2_ atmosphere. Infection was stopped at 0, 16, 24, 48, and 72 h post-inoculation. Oocysts inoculated into wells without HCT-8 cells were used as “background” controls for each time point. Experiments were done at least in triplicate. Control assays with unexcysted and heat-inactivated (95 °C for 15 min) oocysts were performed in conjunction with each series of tests.

The limit of CC–qPCR detection was determined using a dilution series of *C. parvum* parasites at concentrations of 10^4^ to 10^0^ (one) oocysts. Unexcysted oocysts were directly inoculated onto monolayers; the assay was stopped at 48 h post-inoculation. Experiments were performed in triplicate and repeated thrice.

### 2.4. DNA Extraction and Real-Time qPCR

At the end of the incubation period, each well was treated for 1 h at 56 °C with 180 μL of ATL buffer (Qiagen, Hilden, Germany) and 20 μL of proteinase K (Qiagen). DNA isolation was performed according to the instructions for the QIAamp DNA mini kit (Qiagen). Each sample was deposited in duplicate in the qPCR plate. The primers and TaqMan probe used for the real-time PCR were positioned inside a specific 452 bp *C. parvum* sequence reported by Fontaine and Guillot [24]. PCR reactions were carried out in a total volume of 25 μL containing 12.5 µL of iQ™ Supermix (Bio-Rad, Boissy-l’Aillerie, France), primers (forward, 5′-CGCTTCTCTAGCCTTTTCATGA-3′; reverse, 5′-CTTCACGTGTGTTTGCCAAT-3′) and probe (5′ FAM-CCAATCACAGAATCATCAGAATCGACTGGTATC-3′ BHQ1) at 0.4 μM and 0.1 μM (final concentrations), respectively, and 5 μL of DNA template. Assays were run on CFX96™ Thermal Cycler (Bio-Rad) using 96-well hard-shell qPCR plates (BioRad). Cycling conditions were as follows: 95 °C for 3 min, followed by 45 cycles (95 °C for 15 s, 60 °C for min). BioRad CFX Manager Software was used to analyze the amplification curves. The Cq values correspond to the quantification cycle. To evaluate parasite infectivity, the ΔCq was calculated. The ΔCq of a sample is obtained by subtracting the Cq of oocysts inoculated onto HCT-8-free well from the Cq of oocysts inoculated onto HCT-8 monolayer well: ΔCq = (Cqsample_w/o cells_ − Cqsample_w/cells_) for each incubation time. The higher the ∆Cq, the more DNA copies are observed, and thus the more parasitic proliferation there is.

### 2.5. Lamb’s Lettuce: Cryptosporidium Oocysts Spiking, Recovery, and Cell Infection

Fresh lamb’s lettuces (*Valerianela locusta*; Linnaeus, 1753) were purchased as a sealed bag [25]. For sample spiking and optimal recovery, according to Robertson and Gjerde [26], 30 g of lamb’s lettuce leaves was weighed into a homogenizer bag (Bagpage R, Interscience, Saint-Nom-la-Bretèche, France) with a filter porosity <250 μm (Interscience), and 100 μL of spike containing 1 to 10^4^ oocysts was deposited by spotting across the leaves.

Inoculated leaves were incubated for 2–3 h at room temperature, then at 4 °C overnight. After incubation, the oocyst recovery was realized according to the ISO 18744:2016 “Microbiology of the food chain -Detection and enumeration of *Cryptosporidium* and *Giardia* in fresh leafy green vegetables and berry fruits” method with some modifications. The pH of 1 M glycine elution buffer (5.5) was adjusted to 3 for better recovery (data not shown), and for each step, samples were centrifuged at 3000× *g* for 30 min at 4 °C. Before oocyst purification by IMS, pellets were resuspended, and 2 mL of a detergent solution containing 1% SDS (Sigma), 0.1% Tween 80 (Sigma), and 0.001% antifoam Y (Sigma) were added as described by Razakandrainibe et al. [27]. After centrifugation, pellets were resuspended in 7 mL of PBS, and oocysts were recovered by IMS (Isolate, TCS Biosciences, Buckingham, UK). Finally, purified oocysts were directly deposited in wells with and without cells, then they were incubated as described above. After incubation, DNA was extracted as described in Section 2.4. Experiments were performed in triplicate and repeated twice.

### 2.6. Statistical Analysis

Experiments were performed in replicates and repeated at least twice to ensure reproducibility. Data were entered into an Excel spreadsheet (Microsoft, Redmond, WA, USA) to determine the mean and standard deviation. For further analyses, all data were transferred to R version 3.6.3 (R foundation for statistical computing; Vienna, Austria) software. Unless otherwise stated, Student’s t-test analysis was used to determine the statistical significance among the different conditions tested. A *p*-value of less than 0.05 was considered significant.

## 3. Results

### 3.1. Parasitic Proliferation Kinetics Using Excysted and Unexcysted Oocysts Assessed by CC–qPCR

Average Cq values generated from parasites inoculated in HCT-8-free wells were consistently stable (mean: 29.96 ± 0.51; range: 29.19–30.5) with no significant difference (Kruskal-Wallis test *p* = 0.305).

As shown in Figure 1, the peak of ∆Cq was observed 48 h after infection of HCT-8 cells, and the highest ∆Cq was obtained using unexcysted oocysts (∆Cq = 6). At 48 h post-infection, compared to unexcysted oocysts, this study illustrates a decrease of the parasite proliferation (∆Cq = 4.5; *p* = 0.005) with the excysted oocysts (either with NaTC alone or with sodium hypochlorite alone). The combination of sodium hypochlorite and NaTC caused a loss of infectivity of the parasite. Indeed, the ∆Cq observed when using this combination was similar to that of heat-inactivated oocysts.

### 3.2. The Detection Limit of CC–qPCR Method Using Purified Unexcysted Oocysts

The method was evaluated by seeding a known concentration of purified oocysts (ranging from 1 to 10,000). Analytical sensitivity results are shown in Table 1. Detection of *C. parvum* DNA can be achieved with a sensitivity of 100% when inoculating at least 100 oocysts onto HCT-8 monolayers (Table 1). As low as one oocyst could be detected in 27% of inoculated cell replicates (8/30), while only 7% (2/30) of wells without cells were positive.

The results show a linear detection range for *Cryptosporidium* concentration with a limit of 10 oocysts (R^2^ = 0.997) and 100 oocysts (R^2^ = 0.945) inoculated onto HCT-8 monolayers and HTC-8-free wells, respectively (Figure 2).

### 3.3. The Detection Limit of CC–qPCR Method Using Unexcysted Oocysts Recovered from Lamb’s Lettuce

When recovered oocysts from spiked (ranging from 1 to 10,000) lamb’s lettuce were inoculated onto monolayers, the CC–qPCR assay was as sensitive as it was for purified oocysts and enabled the detection of 100 oocysts with 100% sensitivity (Table 2). This method allowed the detection of one spiked oocyst in 25% of the HCT-8 well replicates, which were not detected in the HCT-8-free wells.

A standard curve was established, and the coefficient of determination was calculated. The curve was linear over a range of 10 to 10,000 (R^2^ = 0.992) and 100 to 1000 (R^2^ = 0.91) in wells with and without HCT-8 cells, respectively (Figure 3).

These results suggest that assays should be done in at least quadruplicates to detect as low as one infectious *Cryptosporidium* oocyst.

## 4. Discussion

Cryptosporidiosis outbreaks are on the rise, and numerous foodborne and waterborne outbreaks due to *Cryptosporidium* have been reported [15,28,29]. Detection of *Cryptosporidium* oocysts in a variety of food matrices, including salads, constitutes a threat to human health. Although, since 2016, a standard method is available as the gold standard for the detection and enumeration of *Cryptosporidium* and *Giardia* in/on food matrices, the recovery rate is still low for some foods (<20%) [14]. Thus, rapid and robust methods for detecting infectious parasites in food are urgently needed.

This study aims to propose a method with minimum processing steps to limit potential loss of parasites that combines cell culture and the quantitative PCR (CC–qPCR) to detect infectious *C. parvum* oocysts recovered from leafy greens such as lamb’s lettuce. 

This process relies on the oocysts’ ability to proliferate. The parasite proliferation allows an increase in the number of parasitic DNA copies, which leads to better detection. It also provides information about *Cryptosporidium* infectivity. While mouse infectivity assays have been recognized as the reference method for establishing the ability of oocysts to cause infection, they are expensive, time-consuming, and pose ethical concerns. The combination of cell culture and qPCR techniques provides an alternative to detect infectious *Cryptosporidium* specifically and accurately [20]. Moreover, in vitro cell culture of *C. parvum* has shown equivalent results to those obtained in animals [30]. The human ileocecal adenocarcinoma cell line HCT-8 was used in this study as it was previously found to support the most *C. parvum* infection compared to other cell lines [31,32], has been shown to have a better correlation with the neonate BALB/c and CD-1 mouse models [30,33], and is commonly used for *Cryptosporidium* infectivity assays [22,34,35].

Publications described the association of excystation and cell culture for improving in vitro infection [30,32,36,37,38]; however, to reduce the supplementary loss of parasites, we compared the use of excysted and unexcysted oocysts for infecting monolayers. Sodium hypochlorite and sodium taurocholate are the two compounds that are most frequently used in the literature. In this study, sodium hypochlorite (2.63%) and sodium taurocholate (1.5%) solutions were used alone or in combination. Although these procedures increase the excystation of oocysts (data not shown), our results show reduced parasite proliferation compared to unexcysted oocyst inoculation onto HCT-8 monolayers. Surprisingly, the mean Cq values of inactivated oocysts (95 °C for 15 min) were not significantly different to those of oocysts excysted with sodium hypochlorite plus sodium taurocholate (which resulted in the highest excystation rate), suggesting that this excystation procedure resulted in the inactivation and non-proliferation of the parasite (Figure 1). Sodium hypochlorite helps to thin and remove the outer and inner layers of the oocyst wall and, consequently, weakens the parasites even before their contact with sodium taurocholate [39]. The cytotoxicity of sodium taurocholate for HCT-8 cells has been described [40] and, by extension, may have a lethal effect on the weakened sporozoites.

According to the results of the present study, excystation is not necessary before infecting cells as the parasitic proliferation in HCT-8 cells is decreased. Moreover, excystation is an additional step that can lead to a supplementary loss of parasites, whereas the number of recovered *Cryptosporidium* from food is already low. Thus, the use of unexcysted oocysts for CC–qPCR assays seems to be an interesting and relevant alternative for food analysis and risk assessment. The limit of detection of this CC–qPCR method for purified oocysts or those recovered from parasites spiked in lamb’s lettuce was determined using serial dilutions of *C. parvum* at concentrations of 10^4^ to one oocyst. This method reliably enables the detection of up to 100 infectious oocysts (100% of sensitivity). According to the sensitivity obtained at the lowest doses, the assays should be done in at least quadruplicates to allow the detection of one infectious oocyst in water (3/12) or lamb’s lettuce (8/30) (Table 1 and Table 2).

The present CC–qPCR method allows the assessment of *Cryptosporidium* infectivity in food. The inoculation of parasites in wells with and without cells allows us to calculate the ∆Cq that reflects the parasitic proliferation relative to the *Cryptosporidium* DNA background (in wells without cells). The higher the ∆Cq, the more DNA copies are observed; thus, the more parasitic proliferation there is. However, in a condition when we have the Cq in wells with cells and no signal in wells without cells, the ∆Cq cannot be calculated, but we assume parasites have proliferated. When no signal is observed in wells without cells, the detection limit of classic qPCR is probably reached, and the DNA background cannot be detected. Thus, DNA amplification through parasite proliferation in wells with cells is essential to detect *Cryptosporidium* in the sample. Therefore, the CC–qPCR can circumvent the limits of the classic qPCR and improve the level of detection in complex food matrices compared to direct molecular assays. The results of wells with and without cells are still essential to determine if the observed qPCR signal in wells with cells is a background signal of non-infectious oocysts or due to the proliferation of infectious parasites.

Furthermore, a good correlation is observed between the Cq obtained with and without cells and the quantity of oocysts initially inoculated in wells for 10,000 to 10 (R^2^ = 0.997) and 10,000 to 100 (R^2^ = 0.945), respectively; and of parasites recovered from lamb’s lettuce for 10,000 to 10 (R^2^ = 0.992) and 10,000 to 100 (R^2^ = 0.91), respectively (Figure 2 and Figure 3). These results show that the method can be used to detect, quantitatively, up to 100 infectious oocysts recovered from food and can highlight the parasites’ inactivation (loss of infectivity). Such a performance offers interesting perspectives to assess the efficacy of food processes on oocysts in food.

## 5. Conclusions

The present CC–qPCR method is sensitive, reproducible, and quantitative. It is adequate to successfully detect as low as one infectious *C. parvum* oocysts from lamb’s lettuce if assays are performed in at least quadruplicates. The technique uses unexcysted oocysts and, thus, limits the potential loss of parasites at excystation procedures, washes, and successive centrifugations because the entirety of the sample is analyzed. Furthermore, it provides a valid alternative to the mouse model to evaluate parasite infectivity in food.

This CC–qPCR method constitutes a useful tool to carry out parasite surveillance in food and, thus, to assess consumer exposure to *Cryptosporidium* hazard.

## Figures and Tables

**Figure 1 microorganisms-09-00215-f001:**
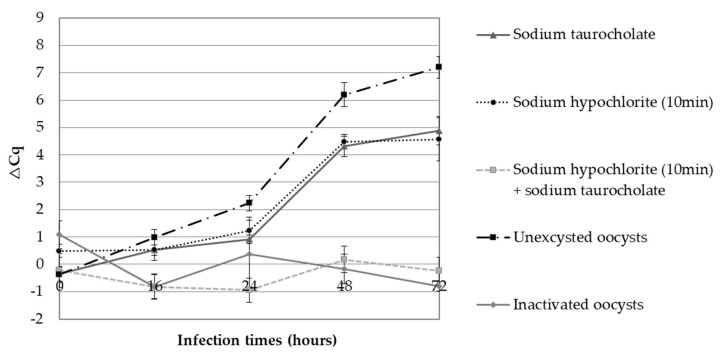
Parasitic proliferation kinetics established from ∆Cq values. Kinetics of the human ileocecal adenocarcinoma cell line (HCT-8) cells infection by excysted oocysts or not. Oocysts excysted with 1.5% sodium taurocholate (grey triangle), with 2.63% sodium hypochlorite (black round), or with the combination of both (gray square). Unexcysted oocysts (black square) and inactivated oocysts (gray diamond). The parasitic proliferation was assessed after DNA purification and quantification of *Cryptosporidium* DNA by qPCR. A positive ∆Cq (∆Cq = Cq_w/out cells_ − Cq_with cells_) indicates that the parasites have multiplied within cells.

**Figure 2 microorganisms-09-00215-f002:**
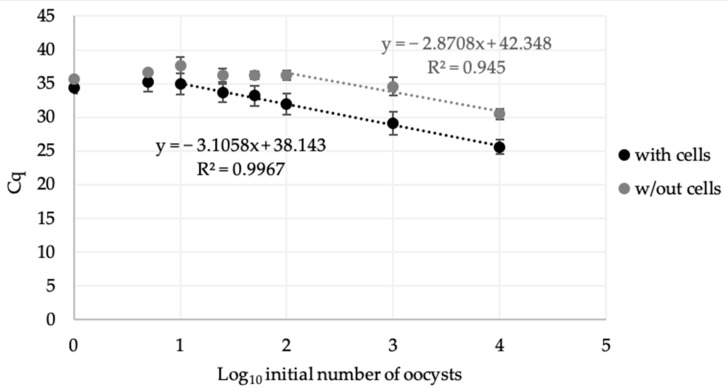
Standard curves of *C. parvum* DNA resulting from CC–qPCR after inoculation of purified oocysts onto wells with (black) or without (grey) HCT-8 cells.

**Figure 3 microorganisms-09-00215-f003:**
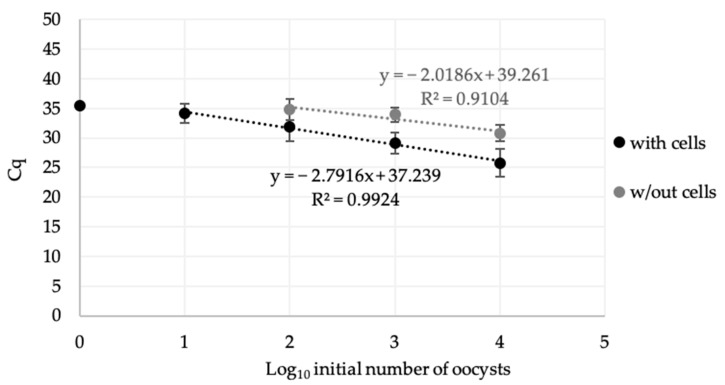
Standard curves of *C. parvum* DNA resulting from CC–qPCR after inoculation of oocysts recovered from lamb’s lettuce onto wells with (black) or without (grey) HCT-8 cells.

**Table 1 microorganisms-09-00215-t001:** Cell culture and quantitative PCR (CC–qPCR) assay results for purified *C. parvum* oocysts.

Number of Inoculated Purified Oocysts		10,000	1000	100	50	25	10	5	1
**Oocysts batch**		2	4	4	3	3	4	3	3
**Cell culture replicates**		9	15	15	12	12	15	12	12
**qPCR repeats**		2	2	2	2	2	2	2	2
**Number of positive wells detected**	**with cells**	18/18	30/30	30/30	22/24	22/24	23/30	5/24	8/30
	**w/out cells**	18/18	29/30	14/30	6/24	7/24	3/30	3/24	2/30
**Cq with cells**		25.55	29.03	31.90	33.16	33.63	34.92	35.18	34.35
**SD with cells**		1.09	1.72	1.57	1.55	1.34	1.58	1.37	0.78
**Cq w/out cells**		30.47	34.54	36.21	36.28	36.24	37.68	36.69	35.63
**SD w/out cells**		0.82	1.36	0.73	0.57	0.98	1.24	0.28	0.00

Grey cases are the results of wells with cells. SD = standard deviation.

**Table 2 microorganisms-09-00215-t002:** CC–qPCR assay results for *C. parvum* oocysts recovered from lamb’s lettuce.

Number of Inoculated Oocysts Per 30 g of Lamb’s Lettuce		10,000	1000	100	10	1
**Oocysts batch**		2	2	2	2	2
**Cell culture replicates**		6	6	6	6	6
**qPCR repeats**		2	2	2	2	2
**Number of positive wells detected**	**with cells**	18/18	12/12	18/18	8/12	3/12
	**w/out cells**	18/18	12/12	9/18	0/12	0/12
**Cq with cells**		25.79	29.16	31.91	34.18	35.40
**SD with cells**		2.38	1.74	2.52	1.58	0.87
**Cq w/out cells**		30.82	33.94	34.86	N/A	N/A
**SD w/out cells**		1.40	1.26	1.77	N/A	N/A

Grey cases are the results of wells with cells. N/A: not available. SD = standard deviation.

## Data Availability

Not applicable.

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
