# Peer review of "Detection of Infectious Cryptosporidium parvum Oocysts from Lamb’s Lettuce: CC–qPCR’s Intake"

_microorganisms, 2021, doi:10.3390/microorganisms9020215_

Round 1

Reviewer 1 Report

Dear Authors,

I am pleased to read the content of interesting results of your publication. Congratulations on the work written clearly to the point and immediately translated into practice.

The purpose of the given review is a method, with a minimum processing steps to limit potential loss of parasites, which combines cell culture and the quantitative PCR (CC-qPCR) for detecting of infectious C. parvum oocysts recovered from leafy green such as lamb’s lettuce. The present CC-qPCR method allows assessment of Cryptosporidium infectivity in food.

The use of un excysted oocysts for CC-qPCR assays seems to be an interesting and relevant alternative for food analyzes and risk assessment. The limit of detection of this CC-qPCR method for purified oocysts or recovered from parasites spiked in lamb’s lettuce was determined using serial dilutions of C. parvum at concentrations of to one oocyst. This method reliably enables detection up to infectious oocysts (100% of sensitivity). This the most important and innovation.  Such performance offers interesting perspectives to assess the efficacy of food processes on oocysts in food such this important Cryptosporidiosis outbreaks are on the rise, numerous foodborne and waterborne outbreaks due to Cryptosporidium have been reported. Especially that detection of Cryptosporidium oocysts in a variety of food matrices, including salads, constitutes a threat to human health.

The paper is scientifically sound and worthy to be considered for publication. The experimental design is correct; the methodologies are correctly described. The statistical analysis is correct.

Best regards

Author Response

We would like to thank the Reviewers and the Editor for careful and thorough reading of the paper and for thoughtful comments and constructive suggestions which helped to improve this manuscript.

Reviewer 2 Report

Very interesting study and of great practical importance. Presented method will certainly useful in the detection of Cryptosportidium.

I only have two suggestions and editorial comments.

  1. Please add full name (author and date when species was published and described) of the Cryptosporidium parvum and Valerianella locusta.

Cryptosporidium parvum Tyzzer, 1907; lamb’s lettuces Valerianella locusta (Linnaeus, 1753). Note: Linnaeus and date are in the parentheses.

  1. Please correct references list according to the Microorganisms: e.g. titles (journal) should be lowercase (e.g. 11, 12, 13 etc); please correct abbreviations, e.g. Int. J. Food Microbiol. (dots); journals must be abbreviated; scientific names must be italicized.
